# Actions Speak Louder Than Words: Health Behaviours and the Literacy of Future Healthcare Professionals

**DOI:** 10.3390/healthcare10091723

**Published:** 2022-09-08

**Authors:** Ewelina Chawłowska, Rafał Staszewski, Agnieszka Zawiejska, Bogusz Giernaś, Jan Domaradzki

**Affiliations:** 1Laboratory of International Health, Department of Preventive Medicine, Poznan University of Medical Sciences, 60-781 Poznań, Poland; 2Department of Hypertension, Angiology and Internal Diseases, Poznan University of Medical Sciences, 61-848 Poznań, Poland; 3Department of Medical Simulation, Poznan University of Medical Sciences, 60-806 Poznań, Poland; 4Department of Social Sciences and Humanities, Poznan University of Medical Sciences, 60-806 Poznań, Poland

**Keywords:** future healthcare professionals, health behaviours, health education, health literacy, health promotion, medical students, Poland, public health workforce, role-models

## Abstract

Our everyday behaviours in life can positively and negatively impact our health, thus cumulatively shaping our lifestyles as more or less healthy. These behaviours are often determined by our knowledge, literacy, motivations and socioeconomic backgrounds. The authors aimed to assess health behaviours and explore variables that may affect persons studying to become future healthcare professionals in Poland. This study was conducted with a group of 275 undergraduate students attending the Poznan University of Medical Sciences representing six different majors of study. We used self-reported, cross-sectional survey conducted through the use of a questionnaire that consisted of one standardised scale (Juczyński’s Health Behaviour Inventory) as well as a self-developed health literacy measure. The students showed average to high levels of health-promoting behaviours (mean HBI = 82.04 ± 11.26). Medium to strong associations were found between these behaviours and high scores on the health literacy scale (*p* = 0.001, r = 0.45 between total scores of the two scales). Dietetics students and female respondents scored significantly better on both scales, which suggests that their self-reported behaviours and health literacy were higher than those of other participants. Exhibiting health-protective behaviours and high health literacy is likely to result in the better individual health of our respondents, but, more importantly, will also influence their future professions. As members of the healthcare workforce they will be responsible for the health of the population and it is crucial for them not only to provide care, education, and guidance, but also to act as role-models for their patients and society.

## 1. Introduction

A few decades ago, people tended to perceive health and illness strictly as biological phenomena. However, the 20th century saw a gradual shift in thinking about health and its determinants. The shift was reflected in a number of milestone public health events and documents such as the 1946 Constitution of the World Health Organization, which defined health as “a state of complete physical, mental and social well-being” [1]. Three decades later, the Alma Ata International Conference on Primary Health Care stressed the importance of wider community efforts in the improvement of health [2]. This in turn contributed to the development of the global strategy *Health for All by the Year 2000*, which advocated for a multisectoral approach to health promotion [3]. Ever since the publication of the Ottawa Charter (1986) and Dahlgren and Whitehead’s *Social Determinants of Health* (1991) it has already been widely recognised that determinants of health cannot be reduced to medical care and traditional public health [4,5,6,7]. Simultaneously, along with the process of industrialisation both Western and non-Western populations have experienced significantly decreased mortality rates and considerable increases in life expectancy. For example, since the year 1950 the global average life expectancy has more than doubled and is now above 70 years. In the United States it increased from 47 years in 1900 to 79 in 2020, in England from 46 to 81, and in Poland from 43 to 76. Similarly, the child mortality rate in the United States declined from 37 deaths per thousand births in 1950 to 6.3 in 2020, in the United Kingdom it dropped from 37.6 to 4.1, and in Poland from 123.9 to 4.4 [8]. A number of studies show that the reduced mortality and extended lifespan of the general population can be attributed to socioeconomic factors and healthy lifestyles [9,10,11,12]. This data, along with the comprehensive and multisectoral approach to health advocated as part of the so-called “new public health”, helped to convince the medical world to recognise a health-promoting lifestyle as an important determinant of one’s health status [13]. Moreover, researchers suggest that the behavioural factors have probably been more important than modern treatments offered by contemporary medicine, as human behaviours affect the risk of getting ill, determine the span and the process of being ill, and can improve the curing process thus contributing to people’s overall well-being [14,15,16,17].

Simultaneously, the burden of communicable diseases declined, and chronic diseases, sometimes referred to as “lifestyle conditions”, have become the predominant cause of morbidity and mortality worldwide [18]. In fact, numerous studies have shown that most non-communicable diseases, including cardiovascular diseases, stroke, diabetes, obesity, chronic respiratory diseases, and some types of cancer, which are the leading causes of death in most developed countries, are strongly related to a number of unhealthy behaviours [19,20]. Individual health outcomes of both communicable and non-communicable diseases are influenced by two kinds of behaviours: risky or unhealthy behaviours (RB, e.g., smoking, alcohol consumption, sedentary lifestyle), and healthy or health-protective behaviours (HB, e.g., healthy diet, physical activity, good sleep, stress-coping strategies, participation in preventive examinations). Unsurprisingly, both kinds of behaviours have become a central focus of public health interventions [21,22].

It often happens that both HBs and RBs adopted during adolescence continue into adulthood [23]. Moreover, the longer RBs persist, the more effort is required to change them [24]. For that reason, the role of healthcare professionals (HCPs) in promoting patients’ awareness of personal risk behaviours and supporting their positive health behaviour change cannot be overestimated, even though such health education efforts are not always successful [25]. A prerequisite to achieving such a behavioural change is possessing sufficient individual health literacy (HL) [26] understood as one’s capacity to access, process, and understand basic health information needed to make appropriate health decisions in everyday life [27,28,29,30]. HCPs can positively influence their patients’ health literacy [31] and one of the factors which can strengthen such influence is the high esteem with which they are held in by the public [32]. Hence, being an authority figure can help a health professional to build trusting relationships with patients and thus to better assist them in positive behavioural change [25,33,34]. It could then be expected that university students should strive to become such role models by adopting healthy attitudes and engaging in HBs.

However, a part of the problem discussed here is that university students, including those at medical universities, are often subjected to various risk factors and environments which can negatively affect their HBs. They are in a dynamic time of transition and need to face not only the demands of their studies, but also often the challenges and hardships of living independently for the first time. As a result, many engage in a wide range of RBs, including lack of sleep, smoking, excessive alcohol consumption, illicit substance use, and unhealthy dietary practices [35]. Thus, even though medical university students often present very good knowledge and health literacy due to the specificity of their studies [36], a stressful and challenging university life may make it difficult for them to apply that same knowledge and literacy to everyday practice [37,38,39,40,41].

Moreover, it was demonstrated that over time, medical students’ involvement in HBs decreased, while their stress levels and burnout rate increased [42]. Many nurses, being well aware that they were also responsible for their patients’ education regarding modifiable health risk factors, did not adhere to healthy lifestyle recommendations [43,44]. A recent study conducted among Israeli medical students and practicing physicians demonstrated that due to work overload and emotional stress, the HBs of healthcare workers declined over time and their health status deteriorated during residency [45]. A study conducted by Mahler et al. (2022) showed that although the majority of Swiss primary care physicians had no or just one of five major lifestyle risk factors, 40% had two or three and 3% had four, with the most prevalent risk factors being physical inactivity, excess weight, and insufficient sleep [46]. Similarly, Lewtak (2008) found that while most Polish family physicians believed that lifestyle was the most important factor influencing individual health, many admitted that their personal lifestyle did not favour health [47].

These findings are significant because medical professionalism requires HCPs to lead not only ethical lives, but also healthy ones. The reason for this is the fact that as knowledgeable and reliable sources of information and advice on various health-related issues, HCPs hold a unique position in promoting and positively influencing healthy lifestyle habits among their patients [48,49,50]. However, since HCPs often fail to adhere to healthy lifestyle recommendations themselves, many patients do not perceive them as role-models. For example, some studies show that physicians’ excessive weight can negatively affect patients’ perceptions of their credibility, level of trust and inclination to follow medical advice [51,52].

Since medical university students will become future HCPs, as public health professionals and health educators they should become health promoting role-models for their patients and demonstrate the very healthy lifestyles they are going to encourage. However, as already mentioned above, students often face various obstacles to translating health-related knowledge into healthy behaviours. In fact, our own students repeatedly mentioned such obstacles during classes. Some of the barriers to a healthy lifestyle listed by the students were poor campus organisation, lack of affordable healthy foods, lack of places to relax, and busy timetables. Incentivised by such observations, we undertook a study aiming to assess the health behaviours and level of health literacy among future Polish HCPs.

## 2. Materials and Methods

### 2.1. Study Design and Data Collection

We undertook a cross-sectional study aimed at examining the health behaviours and literacy of public medical university students. Study participants were recruited from different departments from the Poznan University of Medical Sciences (PUMS) in Poznan, Poland using convenience sampling. Data were collected from 275 undergraduate students of dietetics, radiography, physiotherapy, medicine, nursing and public health.

A self-administered paper-and-pencil survey assessing HB, HL, and selected health and demographic data was carried out. Participation was voluntary and anonymous. Questionnaires were given out to the participants during seminars after the study’s purpose was explained. The respondents had an opportunity to ask questions during the data collection process or cancel their participation and withdraw from the study at any point before the questionnaires were collected. There were no benefits or compensations for participating in the survey.

The survey was conducted after the Bioethics Committee at PUMS confirmed that the study did not constitute a scientific experiment and did not need ethical approval. Informed consent was obtained from all individual students included in the study.

### 2.2. Instruments and Measurements

The questionnaire used in this study comprised the standardised Health Behaviour Inventory (HBI) created by Zygfryd Juczyński [53] and a self-developed 8-item health literacy questionnaire preceded by an ad hoc sociodemographic section designed to collect information such as age, sex, major of study, place of residence, and financial situation. Clinical data such as self-reported body height (cm) and weight (kg) were also collected. Body mass index (BMI, kg/m^2^) was then calculated and divided into 4 categories: underweight (<18.5 kg/m^2^), normal weight (18.5–24.9 kg/m^2^), overweight (25–29.9 kg/m^2^), and obese (≥30 kg/m^2^) [54].

The HBI consists of 24 statements that describe health behaviours and is divided into four subscales: Proper Eating Habits (types of food consumed e.g., whole-grain bread, vegetables, fruits etc.), Preventive Behaviours (adherence to health recommendations and obtaining health information), Healthy Practices (e.g., regular physical activity and getting enough sleep etc.), and Positive Attitudes (e.g., avoiding stressful and depressing situations etc.). Respondents self-determine the frequency of their health-related behaviours over the past year by assigning them points from 1–5 meaning: 1—almost never, 2—rarely, 3—occasionally, 4—often, and 5—almost always. The values of each statement are summed up and the overall indicator of health behaviours is calculated (range between 24–120). Higher scores indicate a greater frequency of health-promoting behaviours. The raw scores are then interpreted as sten scores: 1–4 sten—low scores; 5–6 sten—average scores; 7–10 sten—high scores. The HBI questionnaire is designed for adults and can be used with both healthy and unhealthy individuals. The internal consistency of HBI based on Cronbach’s alpha was 0.85 for the whole instrument and varied from 0.60 to 0.65 for its four subscales. The reliability of the inventory evaluated by the test–retest method was 0.88 [53].

The short self-developed measure for health literacy assessment (HL8) consists of 8 statements related to interactive and critical health literacy [55]. Respondents determine how easy or difficult certain literacy-related activities are by assigning them points from 1 to 4 (1—very difficult, 2—difficult, 3—easy, 4—very easy). The values of each statement are added and the total score for HL is calculated. The maximum possible score is 32 points.

### 2.3. Data Analysis

Statistical analysis of the data was performed with the Polish version of STATISTICA 13 (TIBCO, Palo Alto, CA, USA) and using R version 4.2.1 and R studio version 2022.2.3.492 [56]. Descriptive analysis was carried out and a number of statistical tests were performed to examine the relationships between variables. Depending on the kind of data collected, the following tests were used: Chi-squared test, Mann–Whitney U test, ANOVA Kruskal–Wallis test, and Spearman’s rank correlation coefficient. *p* values < 0.05 were considered significant. Participants’ health behaviours (low, average, high) were shown on the sten scale.

To investigate relationships between the outcomes and explanatory variables in multivariate models, we fitted stepwise multinomial logistic regression models for health literacy outcomes (categorical variables categorised into “very difficult”—a reference category, “difficult”, “easy” and “very easy”), which is a modification of logistic regression used if the dependent variable is a categorical variable with more than two categories, and stepwise linear regression models for the following outcomes: BMI, HBI total, HBI Proper Eating Habits, HBI Positive Attitudes and HBI Healthy Practices, which are continuous variables. The following explanatory categorical variables were introduced into the models: sex (categorised into “female”—a reference category, and “male”), PUMS study programme (categorised into “Dietetics”—a reference category, “Radiography”, “Physiotherapy”, “Medicine”, “Nursing” and “Public Health”), place of residence (categorised into “up to 10,000 inhabitants”—a reference category, “10,000–100,000 inhabitants” and “above 100,000 inhabitants”) and self-reported household’s monthly income per capita (categorised into “below 1000 PLN”—a reference category, “1000–2000 PLN” and “above 2000 PLN”).

## 3. Results

### 3.1. Characteristics of Study Participants

The participating 275 students represented six majors (Table 1). The most numerous major of study was nursing, which comprised 32.4% of the sample. Of the five remaining majors each comprised between 10–15% of the sample. Female students (78%) predominated in the study group. The mean age of respondents was 20.64 years ± 2.28. Almost half of the participants (48.7%) resided in the city of over 100,000 inhabitants. Additionally, 38.9% of respondents reported that their net income per month per one family member was over 1000 PLN (approx. 220 USD), and in 38.5% reported that it was over 2000 PLN (approx. 440 USD). The mean BMI for the entire sample was 21.85 ± 3.11 (21.43 ± 2.73 for female and 23.33 ± 3.84 for male students).

### 3.2. Health Behaviours

The mean total HBI score in the study sample was 82.04 ± 11.26 (Table 2). The mean results from the four HBI subscales ranged between 3.32 and 3.52. While the highest mean results were obtained for Preventive Behaviours (M = 3.52) and Proper Eating Habits (M = 3.49), students reached lower scores in Healthy Practices (3.32).

The total HBI scores were converted to sten scores (Table 3). Although almost a half of the students had medium levels of health-related behaviours (46.5%), nearly one third had low levels (29.1%) and 24.4% reached high levels of healthy behaviours.

Differences between males and females were also examined (Table 4). Female’s total scores were higher than those of males’ (*p* = 0.0190) in two subscales: Proper Eating Habits and Healthy Practices (*p* < 0.05), which may imply that female students’ present healthier behaviours in terms of diet, sleep, and how they spend their leisure time.

Table 5 presents HBI scores for students of each study major. Students of particular majors differed significantly (*p* < 0.05) in terms of their health-related behaviours. The highest total HBI mean scores were found among students of dietetics (88.30 ± 12.92), and the lowest in students of radiography (79.83 ± 10.35). At the same time, no significant differences were found in particular subscales with one exception, the Proper Eating Habits subscale, where mean results were significantly higher among students of dietetics (M = 4.21, SD = 0.48, *p* < 0.0001). This fact corresponds with the special emphasis on diet and nutrition in the dietetics students’ curriculum.

The lack of significant correlations in other subscales may suggest that, irrespective of major, medical university students present similar health behaviours, with the exception of nutritional habits.

### 3.3. Health Literacy

The results of the HL8 are presented in Table 6. The statements with the highest scores were: “If I don’t understand the instructions of healthcare professionals, I ask them for clarification”, “I report any unusual signs and Symptoms to my doctor or other healthcare professional” and also “I talk to healthcare professionals about my health concerns.” It implies that these abilities were considered the easiest for the students.

The statements with the lowest scores were: “I participate in educational programmes on how to take care of my health”, “I ask healthcare professionals for information on how to take care of myself properly” and “When needed, I use counselling or therapy services.” This may suggest that respondents found these competencies most difficult to execute.

Table 7 presents how HL measured by the self-developed eight-item scale correlated with self-reported health behaviours. The correlations were statistically significant (*p* > 0.05) for a majority of statements and occurred in all HBI subscales. The total HBI score positively correlated with every single health literacy question (*p* < 0.0001, r = 0.45), suggesting a link between behaviours and literacy. The Preventive Behaviours subscale showed the strongest association (*p* < 0.0001, r = 0.50) with the total score of the HL questionnaire. The weakest but still significant association (*p* = 0.042) was found in the Healthy Practices subscale.

The Chi-squared and Kruskal–Wallis tests (Table 8 and Table 9) showed statistically significant difference between HL results on the one hand and sex and major on the other hand (*p* = 0.0245 and *p* = 0.0058, respectively). Female respondents scored significantly higher in the HL8. As far as majors were concerned, dietetics and medicine presented higher literacy scores, whereas physiotherapy and radiography scored the lowest in the studied sample.

As presented in Table 9 future dieticians seem to be most literate among the students of different majors included in the study. They are significantly more likely to report unusual symptoms to health professionals, read nutrition labels while buying food and participate in educational programmes than the students of other majors. It may be related to the content of the curriculum and their future role as dieticians (with focus on health education and communication).

### 3.4. Body Mass Index

Body mass index (BMI) was not statistically correlated with either total or subscale scores of the HBI. BMI also did not correlate with health literacy level (*p* = 0.3053) but was associated with sex and major (Appendix A).

### 3.5. Results of the Multivariate Analysis

Although several participants’ characteristics remained statistically significant predictors for these outcomes (usually sex, place of residence and the study programme), these models explain small proportions of the variances of the dependent variables (Appendix A). Moreover, wide 95% confidence intervals indicate low precision of the estimates due to a small cohort size, even if the relationships remain highly significant after controlling for the other explanatory variables. Our analysis also indicates very small but statistically significant differences in health literacy patterns between the students of dietetics and those attending the other programmes. Additionally, we did not find any association between the indices of HL and self-reported financial status. None of the explanatory variables were related to the question two and three of HL8 *(“If I don’t understand the recommendations of healthcare professionals, I ask them for clarification” or “When I disagree with the recommendations of my healthcare professional, I seek a second opinion”*).

Multivariate analysis for BMI (see Appendix A) confirmed that sex and study programme were significant predictors of this composite health outcome: men had significantly higher BMI compared to female students, and students of radiography, nursing and public health, had significantly higher BMI compared to the students of dietetics used as a reference group.

Multivariate analysis identified also several statistically significant predictors for HBI indices (see Appendix A), for HBI total score (male sex and monthly income as negative predictors), for the subscale HBI Proper Eating Habits (male sex, and study programmes other than dietetics as negative predictors, place of living above 100,000 inhabitants as a positive predictor), and for the HBI subscale Healthy Practices (male sex as a negative predictor). None of the explanatory variables was significantly related to the HBI Positive Attitudes subscale.

## 4. Discussion

Although adulthood is the period when one’s healthy lifestyle is being shaped, knowledge about healthy behaviours is not enough as many individuals find it hard to put recommendations into practice [57,58,59,60,61]. Indeed, research shows that although future HCPs are health literate, due to challenges related to stressful and challenging university life and living away from family home, they often find it difficult to apply the knowledge they gain during studies to their everyday lives [37,38,39,40,41]. Consequently, they exhibit a wide range of risky behaviours and unhealthy practices, including poor nutrition, physical inactivity, excessive substance use (e.g., tobacco, alcohol, drugs etc.), poor stress management, and insufficient sleep [37,38,39,40,41]. For example, a study conducted among medical students by Ślusarska et al. (2012) demonstrated that while future physicians presented many health-promoting behaviours (e.g., being a non-smoker, maintaining a proper body weight, and performing physical activity etc.), they still engaged in many unhealthy dietary practices including consuming an inadequate number of meals per day, uneven distribution of meals, inadequate intake of fish, fruits, vegetables and foods rich in fibre; and high salt consumption [62]. Finally, a recent study conducted by Rogowska et al. (2020) demonstrated students’ inadequate lifestyle behaviours regarding diet, psychoactive substance use, coping with stress, physical activity and preventive behaviours [63].

Our study, conducted with the use of Juczyński’s HBI [53], demonstrated that a mean HBI score among PUMS students was average (M = 82.04, with the maximum total score in the Inventory being 120). While the majority reached a medium sten score (46.5%), only one fourth of future HCPs reached a high sten score (24.4%) and one third scored within the low sten (29.1%). Most respondents were characterised as having an average level of Preventive Behaviours, Proper Eating Habits, Positive Attitudes and Healthy Practices. The top-scoring major in our study group was dietetics with students having the highest total HBI score and markedly better results in the Proper Eating Habits scale. This finding supports earlier findings presenting dietetics students as better informed than other students in terms of their nutritional behaviours [64].

For comparison, nursing students from Gdansk, Poland reached a lower mean total HBI score (73.19) than our respondents, with only 16.22% reaching a high sten score, and 39.19% scoring low and 44.59% scoring average. They also scored lower on each HBI subscale [65]. Similarly, a comparative study between nursing students in Poland and Latvia showed lower HBI scores than in our study group (78.08 in Poland and 77.95 in Latvia), with almost a half reaching low stens (49.84% and 44%, respectively), and only about a tenth achieving high stens (13.85% and 10.67%, respectively). Interestingly, they scored lower on all subscales except the Positive Attitudes scale where they had higher results (3.47 and 3.56, respectively) [66].

In a 2020 study conducted by Radosz et al., midwifery, physiotherapy and nursing students from Cracow obtained lower scores than our participants (77.74, 74.31 and 78.12, respectively). Moreover, the majority of students in each group had low total HBI scores (51.1%, 51.5% and 45.5%), and only 16.7% of midwifery students, 11.1% of physiotherapy students and 14.9% of nursing students had high HBI totals [67]. In yet another study conducted by Aleksandra Rogowska (2020) on health behaviours, the average score of physical education students from the city of Opole was also lower than that of students at PUMS and equalled 73.36. Moreover, 51% scored low for health-related behaviours, 36% had average scores, and only 13% presented high levels of healthy behaviours. Again, their subscale scores were lower than those of Poznań students, but their Positive Attitudes scores were similar to those of our participants (3.39) [68]. This result, alongside the findings reported by Marta Mandziuk (2017) [66], may imply that while the overall frequency of pro-health behaviours among PUMS students was relatively higher, their mental health practices fell behind.

Our next focus was health literacy. The HL8 scale used referred to abilities such as communicating with HCPs, adhering to recommendations and instructions, obtaining health information, and seeking health prevention opportunities. An analysis of responses to particular statements in the scale implies that the responding medical university students felt quite confident when communicating with healthcare professionals about their conditions and treatment, but showed some reluctance to utilise health care for preventive rather than for curative purposes and also to deal with any mental health problems. Furthermore, we found strong HBI-HL8 associations, which may mean that health-promoting behaviours among our participants were linked to high HL (i.e., the ability to seek health information, ask health professionals health-related questions, participate in health education programmes, and report health concerns). The strongest link was found between the Preventive Behaviours scores and the total HL8 score. This suggests that high health literacy in our respondents may result in such behaviours such as adherence to health recommendations, seeking health information, and undergoing regular check-ups. The weakest association was found in the Healthy Practices subscale. This might indicate that respondents’ literacy did not always correspond with such behaviours as regular physical activity and getting enough sleep. In general, our research is in line with previous Polish studies which have demonstrated that although most students of medical and health sciences are health literate, the level of their health behaviours is low or, at best, average [69].

Another important finding is that female students not only had higher total HBI scores (82.86 vs. 79.1 in men) but they also exhibited better nutritional habits and healthy practices. This came as no surprise since it is well-established that women tend to be more health literate and engage in health-promoting and preventive behaviours more often than men. For example, Rogowska et al. (2020) showed that Polish female students of physical education drank significantly less alcohol and demonstrated a higher level of overall HB [63]. In addition, more male students were found to be e-cigarette users and smokers [70] and to have more liberal attitudes towards cannabis, including a disregard of its harmful effects and its actual use [71]. Similarly, female students in Japan turned out to have higher health-related responsibility and nutrition scores, but males scored higher on physical activity [37]. Yet other studies showed that while males engaged in physical activity more frequently, reported better general health and declared themselves as happier, they were also more prone to smoking, abusing alcohol and having an unhealthy diet. At the same time, females were reported as having a greater desire to lose weight, eat properly and be more focused on preventive behaviours [72,73,74,75,76,77].

All in all, while it may seem that it is fairly easy for medical university students to maintain a healthy lifestyle, in practice, due to work overload and emotional stress [78], their HBs decline over time and they become even less healthy during their residency. Indeed, research shows that although medical students have good health literacy, during their studies they face many barriers to engaging in HBs, such as having long studying hours [37,38,39,40,41]. Later on, residency brings with it even heavier burdens, and the task of taking care of one’s health becomes much more difficult [45,79]. The available literature suggests that because HCPs often neglect their own health in favour of their professional and personal obligations, their lifestyles tend to be worse than those of the general population [45,80,81,82,83,84,85,86]. For example, in a Polish study conducted by Lewtak (2008), 92% of family physicians defined lifestyle as a key determinant of health, but more than a half admitted that they led unhealthy lifestyles themselves [47]. Similarly, both Cymerys et al. (2009) [87] and Marta Niedźwiedzka and Andrzej Grzybowski (2011) [88] showed that physicians’ dietary behaviours were no better than those of lay persons. Research also indicates that Polish physicians and nurses struggle with problematic alcohol use and with smoking [89,90].

It should be added that a person’s low economic status can present a barrier to incorporating healthy behaviours in one’s lifestyle: satisfying basic needs becomes more important than having enough rest and physical activity, following a balanced diet, or using health services. Polish health expenditure is one of the lowest in EU, and Polish HCPs earn significantly less than their EU counterparts [91]. Lower earnings may not only translate to less healthy behaviours of HCPs, but also to lower motivation to promote health among patients.

This is of key importance because healthcare workers, whose responsibilities include the health promotion and health education of their patients, are expected not only to be aware of lifestyle factors affecting individual health but must also to follow recommendations on preventive behaviours, and thus, become role models in their environments. Research shows that the unhealthy lifestyle of HCPs can negatively impact their patients in a few ways [48,49,50,51,52]. First, it can undermine their health and, consequently, their ability to care for their patients. Second, it can decrease their credibility since patients are more likely to follow health recommendations if they perceive HCPs as role models for pro-health behaviour. Third, HCPs displaying healthy behaviours tend to become more involved in counselling about healthy lifestyle. Finally, HCPs who give strict diet or exercise recommendations to their patients but fail to follow such regimens themselves, may be unable to sympathise with their patients and understand that changing unhealthy habits is not easy. In conclusion, having a healthy lifestyle is important for healthcare workers not only for the sake of their own health, but also for their work performance and for the effectiveness of counselling of patients [48,49,50,51,52,92].

Our analysis also indicates heterogeneity of health literacy among candidates for various professions constituting a modern healthcare system—this supports the need to consider these differences in undergraduate medical education to ensure that all graduates of medical universities present adequate, similar levels in these areas, so that the professionals representing the healthcare system are capable of delivering coherent and consistent messages to its users. Otherwise, there is a risk that patients or other stakeholders will be exposed to disconnected and disordered notions which further weaken the functionality of the healthcare system [93,94].

This study has its limitations which may have an impact on its generalisability and interpretation. First, the rather small sample size and recruitment from one medical university reduces the power of the study and increases the margin of error. Second, the results represent only the opinions of students who agreed to participate in the study and cannot be generalised to the entire student population either in Poznan or in the rest of Poland. Third, non-random sampling prevented a thorough analysis of the socio-demographic, structural and socio-cultural background of the issues discussed in our research. Fourth, as this study is based on the quantitative method only, to gain better insight into students’ health behaviours and literacy, further in-depth studies using qualitative methods would be required.

## 5. Conclusions

Since health behaviours and literacy are fundamental for healthcare professionals as well as for their patients, it is crucial for the institutions where HCPs study, train, and later also work, to incorporate health promotion opportunities into professionals’ lifestyles. Medical universities should ensure that health promotion is present not only in their curricula, but also on their premises.

Unfortunately, our results indicated deficits in health behaviours and literacy among a majority of future healthcare professionals. However, more data are needed to determine the reasons behind this situation and to develop appropriate solutions. While health education is widely available to all medical university students, previous studies demonstrated that there are actual barriers making it difficult for future HCPs to apply health-related knowledge to everyday practice. As we did not investigate the barriers that hinder students’ engagement in HBs, further in-depth research on such factors would be required to guide possible policy changes. Depending on the results of such research, the changes could involve enriching the curricula with evidence-based and up-to-date concepts and examples of good practices regarding health promotion and behaviour change, equipping students with practical skills they could employ in everyday lives, or implementing a healthy settings approach at university premises that would embed health into all aspects of campus culture.

Leading a healthy lifestyle not only contributes to one’s health and productivity but proves to be a motivating and empowering factor for the people around us. Choosing to become a healthcare professional means devoting one’s life to protecting and promoting the health of other people. However, to be successful in this endeavour, an HCP needs to take care of their personal health first and should not forget that actions speak louder than words.

## Figures and Tables

**Table 1 healthcare-10-01723-t001:** Study sample characteristics.

Participants	*n* = 275			*n*	Percent
**Major**	Dietetics			30	10.9%
	Radiography			36	13.1%
	Physiotherapy			42	15.3%
	Medicine			35	12.7%
	Nursing			89	32.4%
	Public Health			43	15.6%
**Sex**	Female			215	78.2%
	Male			60	21.8%
**Age** [years]	(M 20.64 ± 2.28)				
	18–19			139	50.5%
	20–21			62	22.5%
	22–23			39	14.2%
	>24			34	12.4%
**Residence**	Up to 10,000 inhabitants			75	27.3%
	Up to 100,000 inhabitants			66	24.0%
	Above 100,000 inhabitants			134	48.7%
**Net income** per month per one family member	<1000 PLN			41	14.9%
	1000–2000 PLN			107	38.9%
	>2000 PLN			106	38.5%
**Height** [cm]	**All participants** *n* = 275 (170.02 ± 7.82)	**Females** *n* = 215 (167.24 ± 5.81)	**Males** *n* = 60 (179.88 ± 5.84)		
**Weight** [kg]	*n* = 275 (63.42 ± 11.66)	*n* = 215 (60.00 ± 8.54)	*n* = 60 (75.49 ± 13.15)		
**BMI**	(M 21.85 ± 3.11)	(M 21.43 ± 2.73)	(M 23.33 ± 3.84)		
Underweight (<18.5)	*n* = 31 (17.64 ± 0.65)	*n* = 28 (17.66 ± 0.65)	*n* = 3 (17.53 ± 0.73)		
Normal weight (18.5–24.99)	*n* = 194 (21.39 ± 1.67)	*n* = 153 (21.24 ± 1.63)	*n* = 41 (21.93 ± 1.72)		
Overweight (25–29.99)	*n* = 37 (26.39 ± 1.28)	*n* = 27 (26.42 ± 1.41)	*n* = 10 (26.32 ± 0.89)		
Obesity (≥30)	*n* = 5 (32.32 ± 3.72)	-	*n* = 5 (32.32 ± 3.72)		

**Table 2 healthcare-10-01723-t002:** Health Behaviour Inventory scores (*n* = 275).

Subscales and Total HBI	M	SD	95% CI	Median
Proper Eating Habits	3.49	0.74	[3.40, 3.58]	3.55
Preventive Behaviours	3.52	0.64	[3.44, 3.59]	3.50
Positive Attitudes	3.37	0.68	[3.29, 3.45]	3.33
Healthy Practices	3.32	0.32	[3.25, 3.39]	3.33
**Health Behaviour Inventory (total score)**	82.04	11.24	[80.70, 83.38]	82.00

**Table 3 healthcare-10-01723-t003:** Participant HBI scores according to sten norms.

*n*	%	Sten	Score Interpretation	M	SD
80	29.1%	1–4	Low	68.20	5.05
128	46.5%	5–6	Medium	81.70	3.93
67	24.4%	7–10	High	94.33	4.87

**Table 4 healthcare-10-01723-t004:** Sex vs. health behaviours.

	Females *n* = 215 (78.2%)		Males *n* = 60 (21.8%)		*p*
Subscales and Total HBI	M	SD	M	SD	
Proper Eating Habits	3.57	0.73	3.21	0.69	**0.0005**
Preventive Behaviours	3.55	0.65	3.40	0.59	0.0965
Positive Attitudes	3.35	0.69	3.45	0.65	0.3607
Healthy Practices	3.36	0.60	3.16	0.57	0.0313
**Health Behaviour Inventory (total score)**	82.86	11.55	79.10	9.58	**0.0190**

**Table 5 healthcare-10-01723-t005:** Major of study vs. health behaviours.

	Dietetics (*n* = 30) (10.9%)		Radiography (*n* = 36) (13.1%)		Physiotherapy (*n* = 42) (15.3%)		Medicine (*n* = 35) (12.7%)		Nursing (*n* = 89) (32.4%)		Public Health (*n* = 43) (15.6%)		*p*
Subscales and Total HBI	M	SD	M	SD	M	SD	M	SD	M	SD	M	SD	
Proper Eating Habits	4.21	0.48	3.38	0.70	3.48	0.77	3.45	0.68	3.41	0.70	3.27	0.79	**0.0001**
Preventive Behaviours	3.60	0.68	3.41	0.66	3.35	0.63	3.52	0.61	3.59	0.59	3.57	0.72	0.5752
Positive Attitudes	3.59	0.62	3.20	0.69	3.29	0.70	3.55	0.62	3.39	0.72	3.27	0.65	0.0716
Healthy Practices	3.43	0.59	3.37	0.52	3.23	0.46	3.40	0.47	3.33	0.63	3.20	0.80	0.5461
**Health Behaviour Inventory (total score)**	88.30	12.92	79.83	10.35	80.07	9.24	83.57	9.71	82.17	10.78	79.91	13.17	**0.0266**

**Table 6 healthcare-10-01723-t006:** Health literacy questionnaire results.

	Statement	% of Max Score	M	SD
**1.**	I report any unusual signs and symptoms to my doctor or other healthcare professional	70%	2.78	0.92
**2.**	If I don’t understand the instructions of healthcare professionals, I ask them for clarification	80%	3.21	0.88
**3.**	When I disagree with the instructions of my healthcare professional, I seek a second opinion	62%	2.47	0.92
**4.**	I talk to healthcare professionals about my health concerns	69%	2.75	0.93
**5.**	I ask healthcare professionals for information on how to take care of myself properly	53%	2.13	0.89
**6.**	When I buy food, I read nutrition label	64%	2.56	1.02
**7.**	I participate in educational programmes on how to take care of my health	46%	1.83	0.81
**8.**	When needed, I use counselling or therapy services	55%	2.18	0.98
	**Total HL8 score**	**62%**	**19.88**	**4.41**

Note: Answers: very easy, easy, difficult, very difficult.

**Table 7 healthcare-10-01723-t007:** Health literacy vs. health behaviours.

Statement	Proper Eating Habits	Preventive Behaviours	Positive Attitudes	Healthy Practices	Health Behaviour Inventory (total)
I report any unusual signs and symptoms to my doctor or other healthcare professional	***p* = 0.0001** **r = 0.23**	***p* = 0.0001** **r = 0.36**	***p* = 0.0015** **r = 0.19**	***p* = 0.0130** **r = 0.15**	***p* = 0.0001** **r = 0.33**
2. If I don’t understand the instructions of healthcare professionals, I ask them for clarification	*p* = 0.1969 r = 0.08	***p* = 0.0001** **r = 0.30**	***p* = 0.0200** **r = 0.14**	*p* = 0.7515 r = −0.02	***p* = 0.0049** **r = 0.17**
3. When I disagree with the instructions of my healthcare professional, I seek a second opinion	***p* = 0.0001** **r = 0.28**	***p* = 0.0006** **r = 0.21**	*p* = 0.5079 r = 0.04	*p* = 0.5257 r = −0.04	***p* = 0.0095** **r = 0.16**
4. I talk to healthcare professionals about my health concerns	***p* = 0.0007** **r = 0.20**	***p* = 0.0001** **r = 0.37**	***p* = 0.0130** **r = 0.15**	*p* = 0.2100 r = 0.08	***p* = 0.0001** **r = 0.28**
5. I ask healthcare professionals for information on how to take care of myself properly	***p* = 0.0018** **r = 0.19**	***p* = 0.0001** **r = 0.34**	*p* = 0.0733 r = 0.11	*p* = 0.9653 r = −0.01	***p* = 0.0002** **r = 0.23**
6. When I buy food, I read nutrition label	***p* = 0.0001** **r = 0.56**	***p* = 0.0002** **r = 0.22**	***p* = 0.0003** **r = 0.22**	***p* = 0.0124** **r = 0.15**	***p* = 0.0001** **r = 0.41**
7. I participate in educational programmes on how to take care of my health	***p* = 0.0024** **r = 0.18**	***p* = 0.0001** **r = 0.24**	***p* = 0.0272** **r = 0.13**	*p* = 0.1243 r = 0.09	***p* = 0.0002** **r = 0.22**
8. When needed, I use counselling or therapy services	***p* = 0.0001** **r = 0.26**	***p* = 0.0001** **r = 0.38**	***p* = 0.0006** **r = 0.21**	***p* = 0.0226** **r = 0.14**	***p* = 0.0001** **r = 0.34**
**Total HL8 score**	***p* = 0.0001** **r = 0.41**	***p* = 0.0001** **r = 0.50**	***p* = 0.0001** **r = 0.25**	***p* = 0.0425** **r = 0.12**	***p* = 0.0001** **r = 0.45**

**Table 8 healthcare-10-01723-t008:** Health literacy vs. sex.

		Females *n* = 215 (78.2%)			Males *n* = 60 (21.8%)			*p*
	Statement	% of Max Score	M	SD	% of Max Score	M	SD	
**1.**	I report any unusual signs and symptoms to my doctor or other healthcare professional	71%	2.85	0.93	64%	2.57	0.85	**0.0227**
**2.**	If I don’t understand the instructions of healthcare professionals, I ask them for clarification	81%	3.23	0.86	79%	3.15	0.94	0.6368
**3.**	When I disagree with the instructions of my healthcare professional, I seek a second opinion	63%	2.52	0.93	57%	2.28	0.87	0.0632
**4.**	I talk to healthcare professionals about my health concerns	70%	2.81	0.93	63%	2.53	0.91	**0.0282**
**5.**	I ask healthcare professionals for information on how to take care of myself properly	54%	2.15	0.90	51%	2.05	0.85	0.4871
**6.**	When I buy food, I read nutrition label	65%	2.60	0.98	60%	2.40	1.14	0.1839
**7.**	I participate in educational programmes on how to take care of my health	47%	1.86	0.81	43%	1.72	0.78	0.1885
**8.**	When needed, I use counselling or therapy services	56%	2.22	0.99	51%	2.05	0.95	0.2569
	**Total HL8 score**	63%	4.36	20.19	59%	18.75	4.46	**0.0245**

**Table 9 healthcare-10-01723-t009:** Health literacy vs. major.

		Dietetics (*n* = 30) (10.9%)			Radiography (*n* = 36) (13.1%)			Physiotherapy (*n* = 42) (15.3%)			Medicine (*n* = 35) (12.7%)			Nursing (*n* = 89) (32.4%)			Public health (*n* = 43) (15.6%)			*p*
	CONTENT OF THE QUESTION	% of Max Score	M	SD	% of Max Score	M	SD	% of Max Score	M	SD	% of Max Score	M	SD	% of Max Score	M	SD	% of Max Score	M	SD	
**1.**	I report any unusual signs and symptoms to my doctor or other healthcare professional	74%	2.97	0.89	63%	2.53	0.94	64%	2.55	0.89	77%	3.09	0.75	69%	2.78	0.97	72%	2.88	0.88	**0.0445**
**2.**	If I don’t understand the recommendations of healthcare professionals, I ask them for clarification	80%	3.20	0.76	77%	3.08	1.02	80%	3.19	0.77	81%	3.23	0.91	83%	3.31	0.90	78%	3.12	0.88	0.6374
**3.**	When I disagree with the recommendations of my healthcare professional, I seek a second opinion	71%	2.83	0.79	58%	2.33	1.12	57%	2.29	0.86	61%	2.46	0.82	64%	2.55	0.93	59%	2.37	0.87	0.1666
**4.**	I talk to healthcare professionals about my health concerns	73%	2.93	0.91	65%	2.61	0.99	64%	2.55	0.92	73%	2.91	0.89	68%	2.73	0.95	71%	2.84	0.90	0.3688
**5.**	I ask healthcare professionals for information on how to take care of yourself properly	61%	2.43	0.86	53%	2.14	1.07	46%	1.86	0.87	54%	2.14	0.69	54%	2.15	0.92	53%	2.12	0.82	0.1493
**6.**	When I buy food, I read nutrition label	85%	3.40	0.72	59%	2.36	0.93	66%	2.64	0.98	66%	2.63	1.14	58%	2.34	1.07	62%	2.47	0.83	**0.0001**
**7.**	I participate in educational programs on taking care of my own health	60%	2.41	0.87	42%	1.69	0.79	40%	1.62	0.66	46%	1.86	0.77	45%	1.80	0.83	45%	1.81	0.76	**0.0035**
**8.**	When needed, I use counselling or therapy	64%	2.55	0.95	48%	1.92	0.77	47%	1.88	0.83	54%	2.17	0.86	57%	2.30	1.14	56%	2.23	0.97	0.0507
	**Total HL8 Score**	71%	22.57	4.09	58%	18.67	4.97	58%	18.57	4.09	64%	20.40	4.05	62%	19.89	4.40	62%	19.84	4.06	**0.0058**

## Data Availability

The datasets generated and analysed during the current study are available from the corresponding author.

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
