# Peer review of "Actions Speak Louder Than Words: Health Behaviours and the Literacy of Future Healthcare Professionals"

_healthcare, 2022, doi:10.3390/healthcare10091723_

Round 1

Reviewer 1 Report

I have read the content of the article entitled "Actions speak louder than words: Health behaviours and literacy of future healthcare professionals". It is a very interesting, well-written study. However, it is worth verifying some minor punctuation errors.

The article brings new knowledge to the presented research area. The manuscript presents the data clearly and the discussion relates to the results obtained. Perhaps, however, the authors would also be inclined to refer to the issue of the possible impact of specific favorable conditions (teaching organization, campus infrastructure, etc.) or the lack of them, prevailing in the analyzed university, which influenced the obtained results.

Reviewer 3 Report

Important study for health care system- health behaviors have a huge impact on the cost and organization of the system,standard tools, correct statistics.

Please consider describing the cost of earnings in comparison to the EU average.

Please consider the relationship between knowledge and financial possibilities in the conclusions / discussions

Reviewer 4 Report

The article is interesting and can represent a basis for continuing the study in more countries, to increase the degree of generalization of the conclusions; it would also be interesting to compare groups made up of students from the same specialization, for a better accuracy of the results
